# High-Dose Assessment of Transgenic Insect-Resistant Maize Events against Major Lepidopteran Pests in China

**DOI:** 10.3390/plants11223125

**Published:** 2022-11-16

**Authors:** Guoping Li, Tingjie Ji, Shengyuan Zhao, Hongqiang Feng, Kongming Wu

**Affiliations:** 1Key Laboratory of Integrated Pest Management on Crops in Southern Part of Northern China, Ministry of Agriculture and Rural Affairs, Institute of Plant Protection, Henan Academy of Agricultural Sciences, Zhengzhou 450002, China; 2State Key Laboratory for Biology of Plant Diseases and Insect Pests, Institute of Plant Protection, Chinese Academy of Agricultural Sciences, Beijing 100193, China

**Keywords:** transgenic insect-resistant maize, bioassay, target pests, high dose

## Abstract

Lepidopteran pests present a key problem for maize production in China. In order to develop a new strategy for the pest control, the Chinese government has issued safety certificates for insect-resistant transgenic maize, but whether these transformation events can achieve high dose levels to major target pests is still unclear. In this paper, the transformation events of DBN9936 (Bt-Cry1Ab), DBN9936 × DBN9501 (Bt-Cry1Ab + Vip3A), Ruifeng 125 (Bt-Cry1Ab/Cry2Aj), and MIR162 (Bt-Vip3A) were planted in the Huang-huai-hai summer corn region of China to evaluate the lethal effects on major lepidopteran pests, *Spodoptera frugiperda*, *Helicoverpa armigera*, *Ostrinia furnacalis*, *Conogethes punctiferalis*, *Mythimna separata*, *Leucania loreyi*, and *Athetis lepigone,* using an artificial diet containing lyophilized Bt maize tissue at a concentration representing a 25-fold dilution of tissue. The results showed that the corrected mortalities of DBN9936 (Bt-Cry1Ab), DBN9936 × DBN9501 (Bt-Cry1Ab + Vip3A), Ruifeng 125 (Bt-Cry1Ab/Cry2Aj), and MIR162 (Bt-Vip3A) to the seven pests were in the ranges 53.80~100%, 62.98~100%, 57.09~100%, and 41.02~100%, respectively. In summary, the events of DBN9936, DBN9936 × DBN9501, and MIR162 reached high dose levels to *S. frugiperda*. DBN9936 × DBN9501 only at the R1 stage reached a high dose level to *H. armigera*. DBN9936, DBN9936 × DBN9501, and Ruifeng 125, at most growth stages, reached high dose levels to *O. furnacalis,* and these three events at some stages also reached high dose levels to *A. lepigone*. Ruifeng 125 presented a high dose level only to *C. punctiferalis*. However, no transformations reached high dose levels to either *M. separata* or *L. loreyi*. This study provides a support for the breeding of high-dose varieties to different target pests, the combined application of multiple genes and the commercial regional planting of insect-resistant transgenic maize in China.

## 1. Introduction

Genetically modified insect-resistant corn was commercially grown in the United States in 1996 and quickly spread to major corn-producing countries such as Brazil. As a result, pests such as *Ostrinia nublilalis* and *Spodoptera frugiperda* were effectively controlled [1,2], achieving significant economic, social, and ecological benefits [3,4,5]. However, one main threat to sustainable cultivation of transgenic insect-resistant maize is the resistance of target pests. Thus the “high dose/refuge” strategy is crucial for pest resistance management [6]. Its theoretical basis is as follows: (1) insect-resistant crops should express a high dose of insecticidal protein, which can kill almost all resistant heterozygous individuals RS or all sensitive individuals SS; (2) the initial frequency of resistance genes in the target pest population is very low; and (3) adults from resistant crop plots and refuges are randomly mated in the field [6]. Among them, insect-resistant crops expressing high-dose insecticidal proteins, that is, the breeding of insect-resistant varieties with high-dose effects on target pests, is not only the most effective control of field pests but also the most significant key to the effective implementation of resistance management strategy.

Field resistance cases of *S. frugiperda* to the TC1507 event expressing Cry1F and the MON810 event expressing Cry1Ab corn have been reported in Puerto Rico, Argentina, and Brazil [7,8,9]. *Busseola fusca* in South Africa and *Helicoverpa zea* in the United States evolved resistance to MON810 [10,11]. *H. zea* in the United States also presented resistance to MON89034 maize expressing Cry1A.105 + Cry2Ab [12]. A review of the global development and application history of transgenic insect-resistant crops showed that all successful cases strictly implemented the “high dose + refuge” strategy, while others did not [13,14]. Especially when some varieties do not reach high doses, such as the first generation of TC1507 maize expressing single Cry1F, in Puerto Rico, Brazil, and other regions where refuge measures were not in place, resistance problems can easily develop [15]. Thus, the key to resistance management is “source control”, that is, Bt crops planted in the pest occurrence area should have high dose expression of the target pests in the area. High dose expression of Bt crops means that the amount of Bt insecticidal protein expressed by Bt crops can kill 100% of sensitive homozygous individuals (SS) and 95% of sensitive heterozygous individuals (RS) in the target pest population [6], which can be achieved by expressing one or a combination of Bt proteins [16]. It is difficult to directly test RS heterozygous individuals with Bt crops because it is difficult to obtain resistant populations before the application of Bt crops for registration. Therefore, this quantitative index cannot be accurately measured before the development of resistance. Internationally, the dose with an expression level ≥ 25 × LC_99.9_ is usually considered as the standard of operationally high dose, that is, the dose that is 25 times higher than the concentration to kill sensitive larvae [17].

Currently, there are five methods to determine whether transformation events reach high dose levels to target pests [18,19]. (1) The expression level of transformation events to be registered should be determined by enzyme-linked immunosorbent assay (ELISA) or other more reliable techniques, and the expression level should be less than 25 times that of commercially grown varieties before bioassay of registered transformation events. (2) Serial dilution and bioassay were performed on lyophilized Bt crop tissues by artificial diet, and non-Bt crop tissues were used as control. (3) In the common pest occurrence area, a large number of investigations on the occurrence of pests on the plants to be tested for transformation event were conducted to ensure that the expression level of transformation event reached LD_99.99_ or higher, so as to ensure that at least 95% of heterozygous Sr could be killed. (4) A method similar to Method 3 but using controlled infestation with the LD_50_ value of the laboratory pest population had an LD_50_ value similar to that of the field pest population. (5) An instar with a large target pest was found, and the LD_50_ of this instar was 25 times higher than that of the newly hatched larvae. Larvae from this instar were then tested on Bt crops to determine whether 95% or more of the older larvae were killed. A combination of two of these methods is generally recommended to determine that the transformation event has reached a high dose level [20].

Although Bt transgenic insect-resistant maize has not been commercially grown in China, several transformation events have obtained the national production safety certificates. These include the insect resistant and herbicide tolerant maize DBN9936 with *cry1Ab* and *epsps* genes; Ruifeng 125 with *cry1Ab/cry2Aj* and *G10evo-epsps* genes; DBN9501 with *vip3Aa19* and *pat* genes; Zhedaruifeng 8 with *cry1Ab* and *cry2Ab* genes; DBN9936 × DBN9501 (DBN3601T) with *cry1Ab*, *vip3Aa19*, pat, and *mepsps* genes; ND207 with *mcry1Ab* and *mcry2Ab* genes; Bt11 × GA21 with *cry1Ab*, *pat*, and *mepsps* genes; and Bt11 × MIR162 × GA21 with *cry1Ab*, *pat*, *vip3Aa20*, and *mepsps* genes (http://www.moa.gov.cn/ztzl/zjyqwgz/spxx/ accessed on 15 October 2022). Previous studies have shown that DBN9936 [21], DBN3601 [22,23], DBN9936, DBN9501, DBN9936 × DBN9501, Bt11 × MIR162, Ruifeng125 [24,25,26], and other transformation events showed high control effects against the invasive pest *S. frugiperda* and other lepidopteran pests in the laboratory and field, showing good commercialization prospects. In 2021, the Ministry of Agriculture and Rural Affairs of China carried out the pilot work of the commercialization of insect-resistant transgenic maize, and the effect was obvious. It not only effectively prevented and controlled the damage of lepidopteran pests and improved the yield and quality of maize but also reduced the application of insecticides and protected the environment, which indicated that the commercialization of insect-resistant transgenic maize in China is promising.

Maize is the grain crop with the largest sown area in China. The annual planting area is 4.13 × 10^7^ hm^2^, and the total output is 2.6 × 10^8^ t (National Bureau of Statistics, http://data.stas.gov.cn/ accessed on 10 September 2022). It is divided into six production areas, namely North spring corn region, Huang-huai-hai summer corn region, Southwest hilly corn region, South hilly corn region, Northwest inland corn region, and Qingzang plateau corn region [27]. There are great differences in planting area, farming system, and insect species in different maize planting areas. The area of spring sown corn in the North spring corn region accounts for 35% of the total area of corn planted in China. *O. furnacalis* and *Mythimna separata* are the main pests in this area. The Southwest hilly corn region and South hilly corn region accounted for 20%, which are the main planting areas of autumn and winter maize in China, dominated mainly by the newly invaded *S. frugiperda*. *O. furnacalis* is the main pest in Northwest inland corn region, accounting for 4%. The summer maize area in the Huang-huai-hai summer corn region is the largest maize production area in China, accounting for 40% of the planting area. *Helicoverpa armigera*, *O. furnacalis*, *Conogethes punctiferalis*, *Athetis lepigone*, *M. separata*, *Leucania loreyi*, and *S. frugiperda* are common maize pests in this area [15,28]. Therefore, according to the occurrence characteristics of the main pests in different maize growing regions, it is very necessary to breed and plant maize transformation events with high dose expression to the main pests in these areas.

Although we have previously determined that several transformation events have strong insect resistance to *S. frugiperda*, it is not clear whether they have reached high dose levels to *S. frugiperda* and other pests. This is not conducive to seed research and development companies to carry out targeted breeding of insect-resistant transgenic maize varieties and resistance management departments to decide whether to commercialize planting of different transformation in different areas. So to explore whether the transformation events in China Huang-huai-hai summer corn region mainly reached high dose level to lepidopteran pests, an artificial diet containing lyophilized tissues of Bt corn at a 25-fold dilution bioassay was used for research. The study contributes to developing Bt protein expression levels and high dose assessments for several main target pests in China’s existing insect-resistant maize transformation events, which provide technical and theoretical support for the regional layout and resistance management of transgenic insect-resistant maize commercial planting in China.

## 2. Results

Analysis of variance showed that the corrected mortality rates after 7 d and 14 d were related mainly to insect species, different transformation events, and different tissues of transformation events (*p* = 0.000). The corrected mortality rates between 7 d and 14 d were significantly different (*T* = −19.704, *df* = 323, *p* = 0.000), and there was a significant positive correlation (*r* = 0.891, *p* = 0.000). In addition to the direct lethal effect of Bt protein on pests, it has an inhibitory effect on the growth and development of pests so they cannot complete the life cycle. In order to better and more accurately reflect the lethal effect of transformation events on various pests, the corrected mortality rate after 14 days was used as the index to express, as discussed in the following sections.

### 2.1. Corrected Mortalities of DBN9936 (Bt-Cry1Ab) to the Seven Pests

The corrected mortality rates of newly hatched larvae of seven lepidopteran pests on an artificial diet containing lyophilized tissues of DBN9936 (Bt-Cry1Ab) at a 25-fold dilution relative to isogenic negative control are shown in Figure 1, and the corrected mortality rates of 7 d and 14 d after feeding were significantly different (*T* = −9.995, *df* = 80, *p* = 0.000). The 14 d corrected mortality rates of *S. frugiperda* in DBN9936 (Bt-Cry1Ab) at the V6–V8, V12, VT, and R1 stages were 93.43~100%, which were significantly higher than that of R4 (73.72% ± 1.98%) (Figure 1A, *p* < 0.05). The corrected mortality rate of *H. armigera* was 84.79% ± 1.66% in DBN9936 at the V6–V8 stages, which was significantly higher than 58.05~72.70% at other stages (Figure 1B, *p* < 0.05). The corrected mortality rate of *O. furnacalis* in DBN9936 (Bt-Cry1Ab) at the V6–V8, V12, VT, and R1 stages were 100%, which were significantly higher than 94.70% ± 0.76% at the R4 stage (Figure 1C, *p* < 0.05). The corrected mortality rates of *C. punctiferalis* in DBN9936 (Bt-Cry1Ab) at V6–V8 and V12 were 93.59% ± 0.82% and 94.46% ± 0.89%, respectively, which were significantly higher than 88.57% ± 2.03% at R4 (Figure 1D, *p* < 0.05). The corrected mortality rates of *M. separata* of DBN9936 (Bt-Cry1Ab) at V6–V8 and V12 were 87.31% ± 6.43% and 88.35% ± 2.59%, respectively, which were significantly higher than 53.80% ± 3.54% at the R4 stage (Figure 1E, *p* < 0.05). The corrected mortality rates of *L. loreyi* larvae of DBN9936 (Bt-Cry1Ab) at V6–V8, V12, and R4 were 80.06% ± 4.41%, 81.53% ± 3.95%, and 80.13% ± 1.71%, respectively, and the difference was not significant among these stages (Figure 1F, *p* > 0.05). The 14 d corrected mortality rate of *A. lepigone* reached 100% in DBN9936 (Bt-Cry1Ab) at stages V6–V8, V12, and R4, and there was no significant difference (Figure 1G, *p* > 0.05).

### 2.2. Corrected Mortalities of DBN9936 × DBN9501 (Bt-Cry1Ab + Vip3A) to the Seven Pests

The corrected mortality rates of newly hatched larvae of seven lepidopteran pests fed with an artificial diet containing lyophilized tissues of DBN9936 × DBN9501 (Bt-Cry1Ab + Vip3A) at a 25-fold dilution relative to isogenic negative control are shown in Figure 2, and the corrected mortality rates at 7 d and 14 d of feeding were significantly different (*T* = −11.633, *df* = 80, *p* = 0.000). The 14 d corrected mortality rates of *S. frugiperda* of DBN9936 × DBN9501 (Bt-Cry1Ab + Vip3A) at the VT, R1, and R4 stages were 100%, which were significantly higher than 89.36% ± 3.98% and 87.18% ± 0.84% at the V6–V8 and V12 stages, respectively (Figure 2A, *p* < 0.05). The 14 d corrected mortality rate of *H. armigera* of DBN9936 × DBN9501 (Bt-Cry1Ab + Vip3A) at the R1 stage was 100%, which was significantly higher than that of 32.82~82.51% at other stages (Figure 2B, *p* < 0.05). The corrected mortality rates of *O. furnacalis* of DBN9936 × DBN9501 (Bt-Cry1Ab + Vip3A) at V6–V8, V12, VT, and R1 were 100%, which was significantly higher than at the R4 stage with 92.02% ± 2.20% (Figure 2C, *p* < 0.05). The corrected mortality rates of DBN9936 × DBN9501 (Bt-Cry1Ab + Vip3A) at V6–V8, V12, and R4 to *C. punctiferalis* were 84.28% ± 2.37%, 85.92% ± 2.39%, and 88.18% ± 1.36%, respectively, and there was no significant difference among the three stages (Figure 2D, *p* > 0.05). The corrected mortality rates of DBN9936 × DBN9501 (Bt-Cry1Ab + Vip3A) at V6–V8, V12, and R4 to *M. separata* were 64.44% ± 2.61%, 62.98% ± 4.11%, and 66.11% ± 2.29%, respectively, and there was no significant difference among the three stages (Figure 2E, *p* > 0.05). The corrected mortality rates of DBN9936 × DBN9501 (Bt-Cry1Ab + Vip3A) at the V6–V8, V12, and R4 stages to *L. loreyi* were 73.01% ± 3.19%, 63.94% ± 3.98%, and 69.79% ± 2.36%, with no significant difference (Figure 2F, *p* > 0.05). The 14 d corrected mortality rate of DBN9936 × DBN9501 (Bt-Cry1Ab + Vip3A) at the V12 stage to *A. lepigone* was 100%, which was significantly higher than 88.80% ± 1.56% at the V6–V8 stage and 86.30% ± 3.07 at the R4 stage (Figure 2G, *p* < 0.05).

### 2.3. Corrected Mortalities of Ruifeng 125 (Bt-Cry1Ab/Cry2Aj) to the Seven Pests

The corrected mortality rates of newly hatched larvae of seven lepidopteran pests fed with an artificial diet containing lyophilized tissues of Ruifeng 125 (Bt-Cry1Ab/Cry2Aj) at a 25-fold dilution relative to isogenic negative control are shown in Figure 3, and the corrected mortality rates at 7 d and 14 d after feeding were significantly different (*T* = −10.704, *df* = 80, *p* = 0.000). Ruifeng 125 (Bt-Cry1Ab/Cry2Aj) at the V12 stage had the highest corrected mortality rate for *S. frugiperda*, reaching 96.60% ± 0.93%, which was significantly higher than 78.15−89.13% at other stages (Figure 3A, *p* < 0.05). The corrected mortality rate for *H. armigera* of Bt-(Cry1Ab/Cry2Aj) at V6–V8 was 95.35% ± 2.69, which was significantly higher than that of other stages with 57.09~67.26% (Figure 3B, *p* < 0.05). Ruifeng 125 (Bt-Cry1Ab/Cry2Aj) at five growth stages had no significant difference in *O. furnacalis* with 100% corrected mortality (Figure 3C, *p* > 0.05). The corrected mortality rates for *C. punctiferalis* of Ruifeng 125 (Bt-Cry1Ab/Cry2Aj) at V6–V8 and V12 were 100%, which were higher than that of 92.10% ± 2.13% in the R4 stage (Figure 3D, *p* < 0.05). The corrected mortality rates of Ruifeng 125 (Bt-Cry1Ab/Cry2Aj) at the V6–V8 and V12 stages to *M. separata* were 76.87% ± 6.42% and 69.55% ± 4.29%, respectively, which were higher than 59.99% ± 2.13% in the R4 stage (Figure 3E, *p* < 0.05). The corrected mortality rate for *L. loreyi* of Ruifeng 125 (Bt-Cry1Ab/Cry2Aj) at V6–V8 was 88.90% ± 0.67%, which was significantly higher than 75.94% ± 1.73% and 65.92% ± 3.43% in the V12 and R4 stages, respectively (Figure 3F, *p* < 0.05). The 14 d corrected mortality rate of for *A. lepigone* Ruifeng 125 (Bt-Cry1Ab/Cry2Aj) at the V12 stage was 100%, which was significantly higher than 94.39% ± 2.19% and 82.13% ± 0.60% at the V6–V8 stage and R4 stage, respectively (Figure 3G, *p* < 0.05).

### 2.4. Corrected Mortalities of MIR162 (Bt-Vip3A) to the Seven Pests

The corrected mortalities of seven lepidopteran pests on an artificial diet containing lyophilized tissues of MIR162 (Bt-Vip3A) at a 25-fold dilution relative to isogenic negative control are shown in Figure 4. The corrected mortality rates on 7 d and 14 d after feeding were significantly different (*T* = −12.182, *df* = 80, *p* = 0.000). The corrected mortality rates of *S. frugiperda* of MIR162 (Bt-Vip3A) at V6–V8, VT, R1, and R4 were 100%, which were significantly higher than that at V12 (78.81% ± 1.63%) (Figure 4A, *p* < 0.05). The corrected mortality rate of *H. armigera* in MIR162 (Bt-Vip3A) in the R4 stage was 87.20% ± 0.75%, which was significantly higher than that of other stages (41.02~79.51%) (Figure 4B, *p* < 0.05). The corrected mortality rates of *O. furnacalis* in MIR162 (Bt-Vip3A) at the VT and R1 stages were 100%, which were significantly higher than 67.23% ± 8.64%, 80.74% ± 3.30%, and 89.61% ± 0.76% at the V6–V8, V12, and R4 stages, respectively (Figure 4C, *p* < 0.05). The corrected mortality rate of *C. punctiferalis* in MIR162 (Bt-Vip3A) at the R4 stage was 89.04% ± 3.28%, which was significantly higher than that of V6–V8 and V12, 72.26% ± 2.54% and 85.75% ± 2.63%, respectively (Figure 4D, *p* < 0.05). The corrected mortality rates to *M. separata* of MIR162 (Bt-Vip3A) at the V6–V8, V12, and R4 stages were 66.41% ± 4.70%, 79.51% ± 1.67%, and 68.46% ± 4.55%, respectively, and there was no significant difference among the three stages (Figure 4E, *p* > 0.05). The corrected mortality rates of MIR162 (Bt-Vip3A) at V6–V8, V12, and R4 to *L. loreyi* larvae were 75.83% ± 0.83%, 74.71% ± 5.23%, 79.89% ± 3.35%, respectively, and there was no significant difference among the three stages (Figure 4F, *p* > 0.05). The corrected mortality rate of *A. lepigone* in MIR162 (Bt-Vip3A) at the V12 stage was 96.66% ± 1.67%, which was significantly higher than 77.94% ± 1.58% at the V6–V8 stage and 85.46% ± 1.49% at the R4 stage (Figure 4G, *p* < 0.05).

### 2.5. Average Corrected Mortalities of Four Transformation Events to the Seven Pests

DBN9936 (Bt-Cry1Ab) had a significant effect on the average corrected mortality of seven insect species (*F* = 21.166, *df* = 6, 81, *p* = 0.000). The 14 d average corrected mortality rates of *A. lepigone*, *O. furnacalis*, *S. frugiperda*, and *C. punctiferalis* were 100%, 98.94% ± 0.58%, 92.65% ± 2.73%, and 92.21% ± 1.14%, respectively. This was significantly higher than the average corrected mortality of *L. loreyi* (80.57% ± 1.73%), *M. separata* (76.48% ± 6.10%), and *H. armigera* (68.68% ± 2.81%) (Figure 5, *p* < 0.05). The order of lethal effect of DBN9936 (Bt-Cry1Ab) to seven insect species was: *A. lepigone*, *O. furnacalis*, *S. frugiperda*, *C. punctiferalis* > *L. loreyi* ≥ *M. separata*, *H. armigera*.

DBN9936 × DBN9501 (Bt-Cry1Ab + Vip3A) also had a significant effect on the average corrected mortality of seven pests (*F* = 17.004, *df* = 6, 81, *p* = 0.000). The 14 d average corrected mortality rates of *O. furnacalis*, *S. frugiperda*, and *A. lepigone* were 98.40% ± 0.93%, 95.31% ± 1.69%, and 91.70% ± 2.33%, respectively. This was significantly higher than *C. punctiferalis*, *L. loreyi*, *H. armigera*, and *M. separata* with 86.13% ± 1.19%, 68.91% ± 2.10%, 67.99% ± 6.54%, and 64.51% ± 1.62%, respectively (Figure 5, *p* < 0.05). The order of lethal effect of DBN9936 × DBN9501 (Bt-Cry1Ab + Vip3A) to seven insect species was: *O. furnacalis*, *S. frugiperda*, *A. lepigone* ≥ *C. punctiferalis* > *L. loreyi*, *H. armigera*, *M. separata*.

Ruifeng 125 (Bt-Cry1Ab/Cry2Aj) also had a significant effect on the average corrected mortality of seven pests (*F* = 10.419, *df* = 6, 81, *p* = 0.000). The average corrected mortality rates of 14 d were 100%, 97.73% ± 1.45%, and 92.17% ± 2.72% for *O. furnacalis*, *C. punctiferalis*, and *A. lepigone*, respectively. It was significantly higher than the average corrected mortality of *S. frugiperda* (85.44% ± 1.98%), *L. loreyi* (76.92% ± 3.51%), *H. armigera* (69.51% ± 3.89%), and *M. separata* (68.80% ± 3.36%) (Figure 5, *p* < 0.05). The order of lethal effect of Ruifeng 125 (Bt-Cry1Ab/Cry2Aj) to seven insect species was: *O. furnacalis*, *C. punctiferalis*, *A. lepigone* ≥ *S. frugiperda* > *L. loreyi*, *H. armigera*, *M. separata*.

MIR162 (Bt-Vip3A) had a significant effect on the average corrected mortality in 14 d of seven insect species (*F* = 17.505, *df* = 6, 81, *p* = 0.000). The average corrected mortality of *S. frugiperda*, *O. furnacalis*, *A. lepigone*, and *C. punctiferalis* was 95.76% ± 2.28%, 87.52% ± 3.68%, 86.69% ± 2.83%, and 82.35% ± 2.93%, respectively. This was significantly higher than 76.81% ± 1.97% and 71.46% ± 2.82% for *L. loreyi* and *M. separata*, respectively, and 65.53% ± 5.37% for *H. armigera* (Figure 5, *p* < 0.05). The order of lethal effect of MIR162 (Bt-Vip3A) to seven insect species was: *S. frugiperda* ≥ *O. furnacalis*, *A. lepigone*, *C. punctiferalis*, *L. loreyi* ≥ *M. separata*, *H. armigera*.

In general, the lethal effects of DBN9936 (Bt-Cry1Ab), DBN9936 × DBN9501 (Bt-Cry1Ab + Vip3A), Ruifeng 125 (Bt-Cry1Ab/Cry2Aj), and MIR162(Bt-Vip3A) to seven lepidopteran pests were 87.00% ± 1.62%, 83.05% ± 1.99%, 84.46% ± 1.67%, and 81.33% ± 1.79%, respectively, and there was no significant difference among them (*F* = 1.828, df = 3, 323, *p* = 0.142, Figure 5).

## 3. Discussion

The insecticidal effect of insect-resistant transgenic maize exogenous Bt depends on its insecticidal protein expression [29,30]. Previous studies have confirmed that Bt protein expression in different regions, different transgenic crops, and different growth stages of the same transgenic crop has significant spatiotemporal variation [21,31,32,33,34]. For example, the expression level of Cry1Ab in DBN9936 was significantly lower in Xinxiang, Langfang, and Harbin than in Wuhan and Shenyang [21]. Similarly, the expression level of Cry1Ab in MON810 maize differed 20-fold, on average, in different regions [35]. This variation may expose local target pests to low and medium dose levels, which not only affects their field control effectiveness but also increases their survival rate due to exposure to sublethal doses, accelerating the evolution of resistance [36]. Therefore, it is of great significance for field planting layout and resistance management techniques for specific transformation events to determine whether insect-resistant transgenic crops achieve high dose levels to major pests in the local area.

Artificial diet containing lyophilized tissues of Bt crop events at 25-fold dilution bioassay is one of the most commonly used high-dose assays. We used this method to determine the high-dose levels of four transformation events to different lepidopteran pests at different stages. The results showed that two transformation events, DBN9936 × DBN9501 (Bt-Cry1Ab + Vip3A) and MIR162 (Bt-Vip3A), reached high dose levels to *S. frugiperda*, and DBN9936 (Bt-Cry1Ab) approached high dose levels. This is consistent with the results of this study [37]. The lethal sensitivity of *S. frugiperda* population to five Bt proteins in Yunnan was Vip3Aa > Cry1Ab > Cry1F > Cry2Ab > Cry1Ac. Therefore, planting insect-resistant maize expressing Cry1Ab, Vip3Aa, or superimposed Cry1Ab + Vip3Aa can meet the requirement of high dose of *S. frugiperda*. The corrected mortality of DBN9936 (Bt-Cry1Ab), DBN9936 × DBN9501 (Bt-Cry1Ab + Vip3A), and Ruifeng 125 (Bt-Cry1Ab/Cry2Aj) to *O. furnacalis* was more than 99.99%, while MIR162 (Bt-Vip3A) reached 100% only at the R1 and R4 stages. Our previous laboratory bioassay showed that *O. furnacalis* exhibited a high sensitivity to Cry1Ab with a LC_50_ value of 2.11 (1.64–2.19) ng/cm^2^, while its sensitivity to Vip3A was low with a value 328.44 (183.99–660.54) ng/cm^2^ [38], which is a 155-fold difference. Studies have shown that Vip3A has little or no activity on *O. nubilalis*, which may be caused by the two different species. Therefore, planting Cry1Ab-based multi-gene superimposed pest resistant maize is suggested to meet the high dose demand of *O. furnacalis*. In the previous study, the sensitivity of *C. punctiferalis* to different Bt proteins was similar to that of *O. furnacalis* [38]. However, in this study, only Ruifeng 125 (Bt-Cry1Ab/Cry2Aj) was exposed to high dose levels under the determination of 25-fold dilution concentration. The reason for this result needs to be studied further.

DBN9936 × DBN9501 (Bt-Cry1Ab + Vip3A) was close to the high dose level to *H. armigera*, while DBN9936 (Bt-Cry1Ab), Ruifeng 125 (Bt-Cry1Ab/Cry2Aj), and MIR162 (Bt-Vip3A) did not reach the high dose level to *H. armigera*. Compared with other pests, *H. armigera* was the most sensitive to Cry2Ab [38]. It is suggested that multi-gene superimposed insect-resistant maize with Cry1Ab + Vip3A and Cry2Ab + Vip3A can meet the demand of high dose of *H. armigera*. The four transformation events did not reach high dose levels to *M. separata* and *L. loreyi*, which is consistent with our study that their sensitivity to Cry1Ab and Vip3A is lower than that of *H. armigera* and *O. furnacalis* [38]. Therefore, multigene superposition is more essential for *M. separata* control to achieve high dose levels. High dose levels of DBN9936 (Bt-Cry1Ab) were reached to *A. lepigone*, and near high dose levels of DBN9936 × DBN9501 (Bt-Cry1Ab + Vip3A) and Ruifeng 125 (Bt-Cry1Ab/Cry2Aj) were reached to this pest as well. *A. lepigone* feeds mainly on the base of maize stems, but in this paper, leaf, silk, and grain were used to determine the high dose levels of the four transformation events, which may not accurately reflect the actual situation.

“High dose/refuge” strategy is an effective measure to ensure sustainable application of insect-resistant transgenic crops [13,39,40,41]. Learning from foreign transgenic insect-resistant corn and resistance of target insect pest management experience and lessons, on the basis of the biological characteristics of major pests in maize growing areas of China, such as regional occurrence, host infestation and migration and dispersal, we proposed a planting layout of transgenic insect-resistant maize with “zoning layout and source control” and a high-dose/refuge resistance management strategy suitable for China’s national conditions, and effectively implemented the refuge strategy [15]. In terms of the layout of transgenic insect-resistant maize, the autumn and winter corn in Southwest hilly and South hilly corn region in China are the annual breeding areas of *S. frugiperda* and *M. separata*, the concentrated landing sites of imported populations from abroad, and the important insect sources in the Huang-huai-Hai summer corn region and North spring corn region of China. Therefore, in order to reduce the occurrence of *S. frugiperda* and *M. separata* in the source area, Bt maize varieties should be planted to efficiently control *S. frugiperda* and *M. separata* and meet the requirements of high dose. The Huang-huai-hai summer corn region is the main occurrence area of *H. armigera*, *O. furnacalis*, *C. punctiferalis*, and *A. lepigone*. Bt maize varieties are planted to meet the requirements of high-dose control of *H. armigera*, *O. furnacalis*, *C. punctiferalis* and *A. lepigone*. In the Northern spring corn region, cultivars with high dose requirements for *O. furnacalis* are planted.

According to the results of high-dose determination of different transformation events against different pests, the research and application of transgenic insect-resistant maize in China should focus on the following points. For areas dominated by a single pest of *S. frugiperda*, Bt maize events such as Cry1Ab + Vip3Aa, expressing Vip3Aa or superimposed with Cry1Ab, should be cultivated. In the area where *S. frugiperda* and *M. separata* co-occur, the combined application of Cry1Ab + Vip3Aa, Cry1F, and Cry2Ab can also achieve an effective high dose level for *M. separata* [38]. For the co-occurrence areas of *H. armigera*, *O. furnacalis*, *C. punctiferalis*, and *A. lepigone*, multi-gene insect-resistant maize varieties containing mainly Cry1Ab + Cry2Ab or combined with Vip3A should be cultivated to achieve the goal of controlling multiple target pests. For the areas where the prevention and control of *O. furnacalis* is the main goal, we should focus on breeding varieties containing Cry1Ab and Cry1Ab + Cry2Aj, which can effectively control the damage of *O. furnacalis*. At the same time, according to the occurrence and damage characteristics of pests, the type and size of refuge should be formulated for each event in each area. Because China has commercially planted Bt-Cry1Ac Cotton since 1997 [39], one concern arises for cross-resistant pest development with Bt corns. This is not an important issue because more than 80 percent of cotton is grown in western China, which is a non-corn-producing area. For mixed cotton- and maize-growing areas in eastern China, the planting of Bt-Cry1A maize with cross-resistance together with the cotton should be avoided in consideration of resistance management.

Clarifying the relationship between the dose expression of Bt-gene-resistant crop toxin protein and the pest response to it is one of the important aspects of the target pest resistance management work, which is conducive to the establishment of resistance management measures, such as the establishment and size of structural refuge or seed mixed refuge [42,43]. In the United States and Canada, according to the standards proposed by the USEPA, the only transformation with a clear evaluation of whether the transformation meets the high dose standard for a few pests are the following: Bt11 and MON810 expressing Cry1Ab did not reach high dose levels to *H. zea* and *S. frugiperda* but showed high dose levels to *O. nubilalis* [44]. Cry1A.105 and Cry2Ab in MON89034 single protein did not reach high doses to *H. zea* and *O. nubilalis* and was unknown to *S. frugiperda*. However, MON89034 showed better field resistance to these three pests than MON810 [45]. TC1507 expressing Cry1F was not high dose to *H. zea* but was high dose to *O. nubilalis* and was unknown to *S. frugiperda* [44]. MIR162 expressing Vip3A did not reach high dose to *H. zea* and *O. nubilalis* but reached high dose to *S. frugiperda* [46]. As for other pests, such as *Agrotis ipsilon*, *Diatraea grandiosella*, *D. saccharalis*, and *S. exigiua*, corresponding high-dose data are lacking, although they are active [36]. Most of the studies focused on insect resistance after laboratory and field planting, so they could not give clear high-dose results, and it was expressed as close to high dose [47], low dose [48], moderate dose [49], and other designations. Among the few results of the above definitive evaluation, high-dose expression assay was used by 25-fold dilution of lyophilized tissues of transgenic insect-resistant crops. This method—the transgenic insect-resistant crops lyophilized tissues 25 times dilution method—is easy to implement and also the most direct support as it reached 25 times higher doses of the evaluation method. We used this method before China’s commercial cultivation of different transformations not only to the *S. frugiperda*, *H. armigera*, and *O. nubilalis* main pests (such as the high dose assessment) but also carried it out for other four insect pests, namely *M. separata*, *L. loreyi*, *A. lepigone*, and *C. punctiferalis*. The high-dose levels of different transformations to major maize pests in China were comprehensively and systematically evaluated, which is of great significance for guiding the breeding and regional commercial planting of insect-resistant transgenic maize varieties in China.

In this study, the high-dose levels of four transformation events on seven major lepidopteran pests in China were measured, and the dose–response relationships between different tissues of different transformation events and different pests were preliminarily clarified, which provided a basis for the breeding and application of transgenic maize varieties and the establishment of refuges in China. However, as a result of different regions and varieties of genetically modified crops, Bt protein expression at different stages of genetically modified crops have significant differences in characteristics of space and time. For the future, further study is needed on the harm of the pests on corn as well as feeding characteristics in different regions of China’s corn belt. Resistance monitoring is the basis for resistance management of the target pests. The susceptible baselines and resistance allele frequencies of the major target pests in different ecological regions should be established before commercialization, and a regular program for resistance monitoring should be conducted after commercialization.

## 4. Materials and Methods

### 4.1. The Transformation Events of Insect-Resistant Transgenic Maize

DBN9936 transformation event maize (Bt-Cry1Ab), DBN9936 × DBN9501 event (Bt-Cry1Ab + Vip3A), and isogenic negative control (Nonghua 106) were provided by Beijing DaBeiNong Biotechnology Co., Ltd. (Beijing, China). Ruifeng 125 transformation event (Bt-Cry1Ab/Cry2Aj) and its isogenic negative control (Hongshuo 899) were provided by Hangzhou Ruifeng Biotechnology Co., Ltd. (Hangzhou, China). MIR162 transformation event (Bt-Vip3Aa) and its isogenic negative control (Xianda 901) were provided by Syngenta Biotechnology (China) Co., Ltd. (Beijing, China). MIR162 transformation event (Bt-Vip3Aa) has been commercially grown in the United States, Brazil, and other countries. All the above maize varieties were planted in the transgenic maize field (35°13′ N, 113°42′ E) of the Modern Agricultural Science and Technology Base of Henan Academy of Agricultural Sciences located in the Huang-huai-hai summer maize region on 25 June 2020, with an area of 200 m^2^ in each plot. Plant spacing was 28 cm, row spacing was 60 cm, and spacing between plots was 1.5 m, repeated three times; routine water and fertilizer management were implemented.

When maize plants grew to the V6–V8, V12, VT, R1, and R4 stages, tissue samples of leaves, tassels, silk, and grains were taken as the criterion, as shown in Table 1. After each sampling, the Cry1Ab or Vip3A protein expression was confirmed by dipstick tests (AA0331-LS for testing Cry1Ab and AA1632-LS for testing Vip3A, Shanghai Youlong Biotechnology Co., Ltd. (Shanghai, China)), and samples were stored on dry ice and stored in −20 °C freezer within 2 to 4 h. The fresh tissue was ground into a fine powder by a low-temperature pulverization mixer (Robot coupe model R10 V.V.SV, speed 3000 rpm) and then dried by a freeze dryer (Ningbo Xinzhi Biotechnology Co., Ltd. (Ningbo, China), Scientz-12nd) at −50 °C for 24 h. After drying, they were divided into 50 mL centrifuge tubes and stored in the refrigerator at −80 °C until use.

### 4.2. Collection and Culture of Insect Species

The susceptible strain of *S. frugiperda* was collected in a maize field of Mengmao Town, Ruili, Dehong Prefecture, Yunnan Province, China, in January 2019. The collected insect species were mainly the 3rd to 5th instar larvae. The population was sensitive to Bt after laboratory biological tests [37]. *H. armigera, O. furnacalis, A. lepigone*, and *C. punctiferalis* were collected from Modern Agricultural Science and Technology Base of Henan Academy of Agricultural Sciences (Yuanyang County, Henan Province) from 2015 to 2016. *H. armigera* and *O. furnacalis* were collected from conventional corn ears and *C. punctiferalis* was collected from both corn and sorghum. *A. lepigone* was captured by light trap. *M. separata* population was collected from the maize field in Lingbao County, Henan Province in 2016, and the larvae were collected as 4–5 instar larvae. The *L. loreyi* larvae were collected from the spring maize field in Ganan Town, Pingqiao District, Xinyang City, Henan Province in 2019.

*S. frugiperda*, *H. armigera*, *O. furnacalis*, *M. separata*, *L. loreyi*, and *A. lepigone* larvae feed formula in laboratory, artificial ingredients with corn flour, soybean meal, wheat germ and bran, casein as the main ingredient [50] and *C. punctiferalis* feed with chestnut powder and corn flour were acquired, and the formula is shown in [51]. Adults were fed 5–10% honey water to supplement nutrition and water in the cage (40 × 30 × 25 cm^3^). *O. furnacalis* eggs were collected using wax paper, *S. frugiperda*, *H. armigera, C. punctiferalis*, and *A. lepigone* eggs were collected using white medical gauze. *M. separata* and *L. loreyi* eggs were collected using nylon rope.

All larvae and adults of the above species were incubated in an incubator with temperature of (27 ± 1) °C, humidity of 60% ± 10%, and photoperiod of L/D = 16 h/8 h. No chemical insecticides or Bt insecticidal proteins were exposed in the feeding process of the test insects. The specific information of the insect source is shown in Table 2.

### 4.3. High-Dose Bioassays

Dilution and bioassay of lyophilized Bt maize tissues were performed using artificial diet high-dose assays [52,53]. When artificial diets cooled ca. 50 °C, 0 g and 20 g of transgenic insect-resistant maize tissue lyophilized powder and 20 g and 0 g of isotype control maize tissue lyophilized powder were mixed into 480 g artificial diet and stirred evenly, and the concentrations of 0 (0 times) and 4% (25 times) were finally formed. After the artificial diets solidified, 1 m^3^ artificial diet portions were put into 128-well culture plates (diameter of each well 16 mm; height: 13 mm), and one neonate larvae (0–24 h old) of each species was added to each well using a fine brush, and then covered with a breathable plastic mucous membrane to prevent larvae from escaping. The bioassays were repeated three times for each species, forty-eight larvae in each concentration for a total of 144 larvae. All culture plates were incubated at 27 ± 1 °C, 60–70% relative humidity, and 16 h/8 h light. The number of dead larvae was recorded at 7 d and 14 d, respectively. The larvae that could not move normally were considered dead, and the larvae that did not reach the second instar after 7 d and 14 d were also considered dead.

### 4.4. Statistics and Analysis

Equations (1) and (2) were used to calculate the mortality rate and corrected mortality rate of several pests in different stages of different maize transformation events. If the corrected mortality rate reached 100% under the treatment of 4% concentration (25 times dilution concentration), it indicated that the Bt protein content of the maize transformation event in this stage reached the requirement of high dose for the pests.
Mortality (%) = Number of dead insects after treatment/total number of insects tested before treatment × 100%(1)
Corrected mortality (%) = (treatment group mortality − control group mortality)/(100 − control group mortality) × 100%(2)

The differences were analyzed by one-way analysis of variance. Duncan’s new complex range method was used for significance test. SPSS 20.0 software was used for statistical analysis of the test data.

## 5. Conclusions

Herein, we tested the high-dose levels of four transformation events on seven major lepidopteran pests in China. Different transformations at different growth stages showed different dose–mortality relationships for different pests. The three events of DBN9936 (R1), DBN9936 × DBN9501 (VT, R1, R4), and MIR162 (V6–V8, VT, R1, R4) reached high dose level to *S. frugiperda*. DBN9936 × DBN9501 (R1) reached a high dose level to *H. armigera*. DBN9936 (V6–V8, V12, VT, R1), DBN9936 × DBN9501 (V6–V8, V12, VT, R1), and Ruifeng 125 (V6–V8, V12, VT, R1, R4) reached high dose levels to *O. furnacalis.* DBN9936 (V6–V8, V12, R4), DBN9936 × DBN9501 (V12), and Ruifeng 125 (V12) reached high dose levels to *A. lepigone*. Ruifeng 125 (V6–V8, V12) reached a high dose level to *C. punctiferalis*. No transformations reached high dose levels to either *M. separata* or *L. loreyi*. The results of this study can provide support for the breeding of high-dose varieties for different target pests, the combined application of multiple genes, and the commercial regional planting of insect-resistant transgenic maize in China.

## Figures and Tables

**Figure 1 plants-11-03125-f001:**
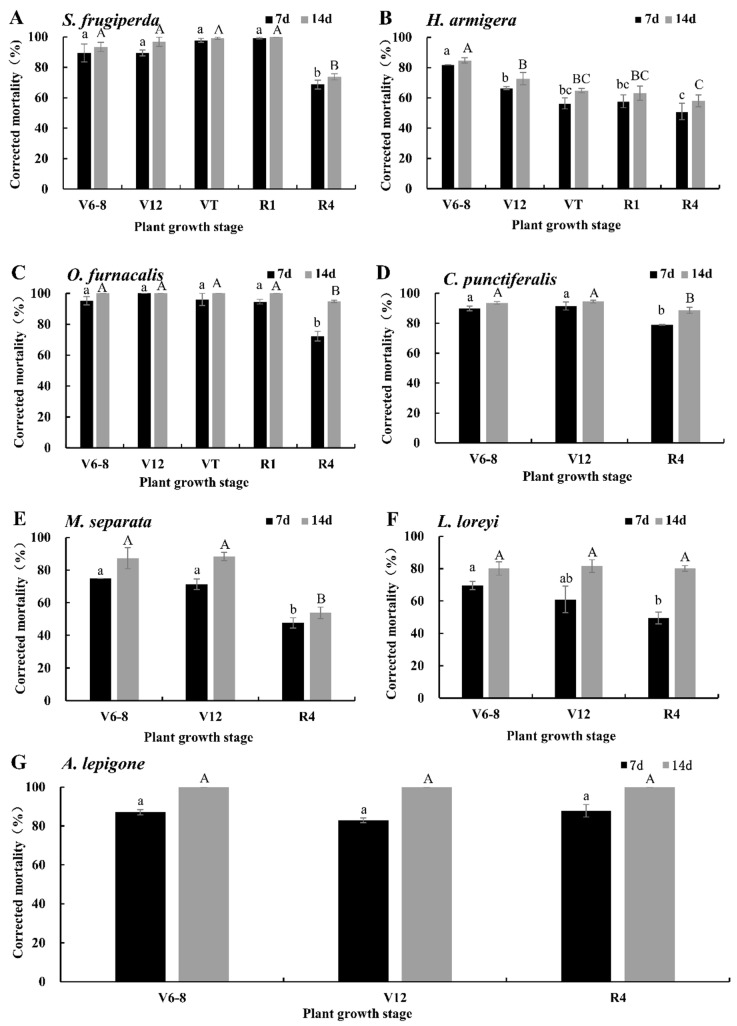
Corrected mortalities of neonates for seven lepidopteran pests on artificial diet containing lyophilized different tissues of DBN9936 corn (expressing Cry1Ab) in different growth stages at a 25-fold dilution relative to the isogenic negative control. Values represent means ± SE. Different lowercase and uppercase letters above black and gray bars indicate significant difference for the same treatment time by Duncan’s multiple range test (*p* < 0.05). (**A**) *S. frugiperda*; (**B**) *H. armigera*; (**C**) *O. furnacalis*; (**D**) *C. punctiferalis*; (**E**) *M. separata*; (**F**) *L. loreyi*; and (**G**) *A. lepigone*.

**Figure 2 plants-11-03125-f002:**
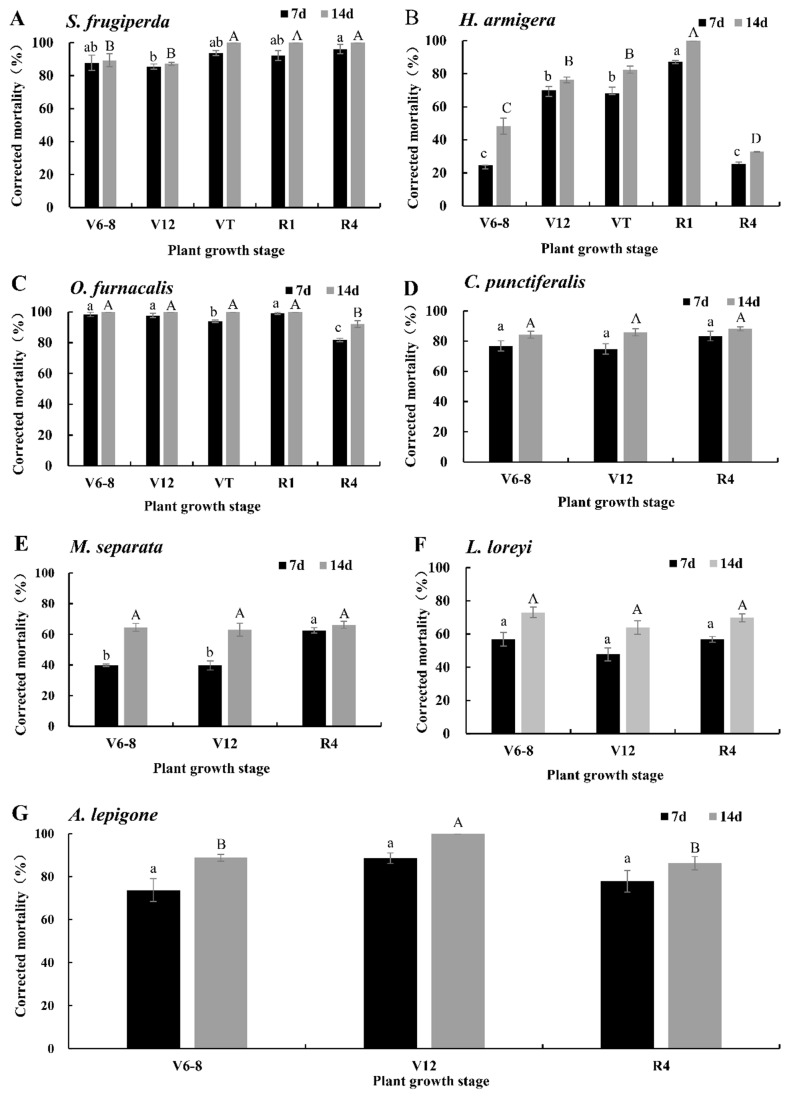
Corrected mortalities of neonates for seven lepidopteran pests on artificial diet containing lyophilized different tissues of DBN9936 × DBN9501 corn (expressing Cry1Ab and Vip3Aa) in different growth stages at a 25-fold dilution relative to the isogenic negative control. Values represent means ± SE. Different lowercase and uppercase letters above black and gray bars indicate significant difference for the same treatment time by Duncan’s multiple range test (*p* < 0.05). (**A**) *S. frugiperda*; (**B**) *H. armigera*; (**C**) *O. furnacalis*; (**D**) *C. punctiferalis*; (**E**) *M. separata*; (**F**) *L. loreyi*; and (**G**) *A. lepigone*.

**Figure 3 plants-11-03125-f003:**
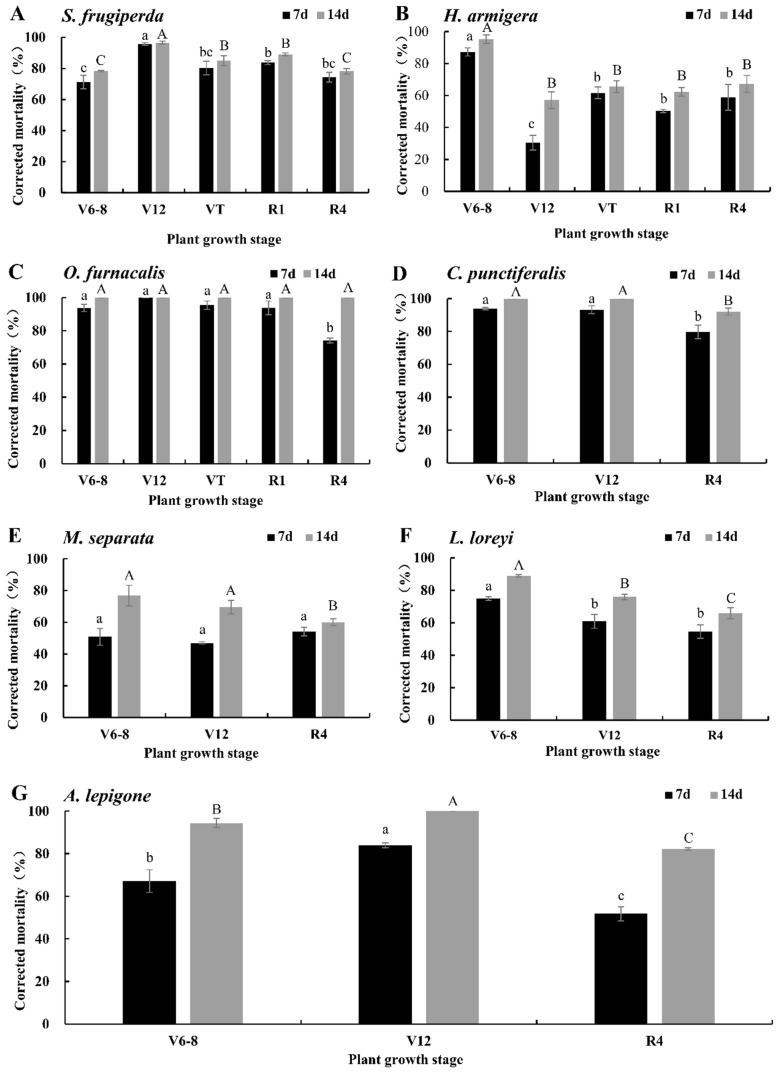
Corrected mortalities of neonates for seven lepidopteran pests on artificial diet containing lyophilized different tissues of Ruifeng 125 corn (expressing Cry1Ab/Cry2Aj) in different growth stages at a 25-fold dilution relative to the isogenic negative control. Values represent means ± SE. Different lowercase and uppercase letters above black and gray bars indicate significant difference for the same treatment time by Duncan’s multiple range test (*p* < 0.05). (**A**) *S. frugiperda*; (**B**) *H. armigera*; (**C**) *O. furnacalis*; (**D**) *C. punctiferalis*; (**E**) *M. separata*; (**F**) *L. loreyi*; and (**G**) *A. lepigone*.

**Figure 4 plants-11-03125-f004:**
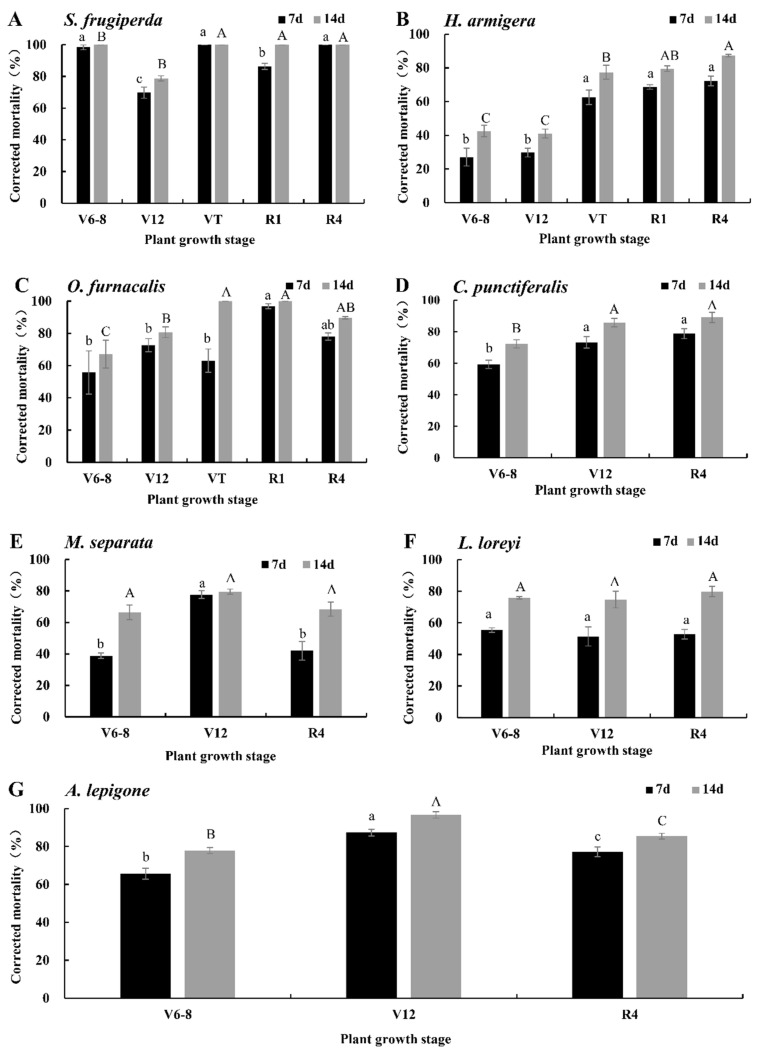
Corrected mortalities of neonates for seven lepidopteran pests on artificial diet containing lyophilized different tissues of MIR162 corn (expressing Vip3Aa) in different growth stages at a 25-fold dilution relative to the isogenic negative control. Values represent means ± SE. Different lowercase and uppercase letters above black and gray bars indicate significant difference for the same treatment time by Duncan’s multiple range test (*p* < 0.05). (**A**) *S. frugiperda*; (**B**) *H. armigera*; (**C**) *O. furnacalis*; (**D**) *C. punctiferalis*; (**E**) *M. separata*; (**F**) *L. loreyi*; and (**G**) *A. lepigone*.

**Figure 5 plants-11-03125-f005:**
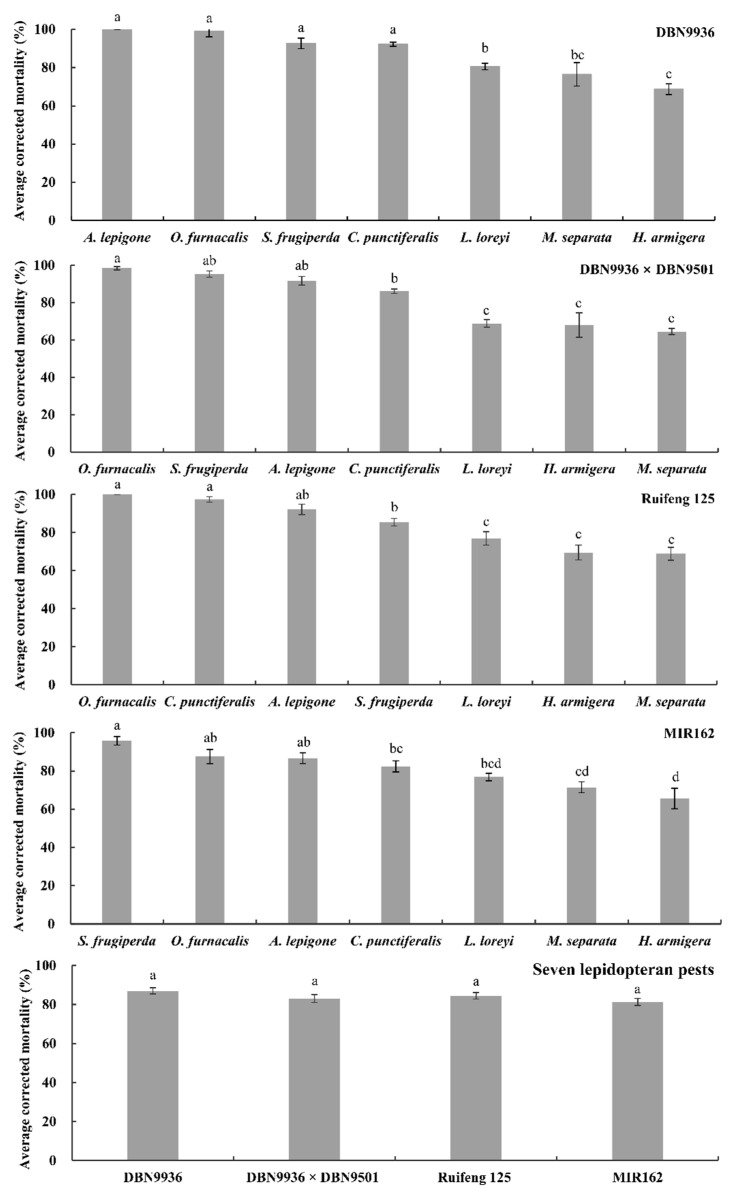
Average corrected mortalities of neonates for seven lepidopteran pests on artificial diet containing lyophilized different tissues of insect-resistant maize transformation in different growth stages at a 25-fold dilution relative to the isogenic negative control. Values represent means ± SE. Different lowercase letters above bars indicate significant difference by Duncan’s multiple range test (*p* < 0.05).

**Table 1 plants-11-03125-t001:** Maize at different growth stages and tissue sampling requirements.

Maize Growth Stage	Tissue	Specific Sampling Requirements
V6–V8 (6–8 leaves have visible collars)	Leaf	The youngest leaf that emerged and was at least 20 cm in length was cut from the leaf tip.
V12 (12 leaves have visible collars)	Leaf	The youngest leaf that emerged and was at least 20 cm in length was cut from the leaf tip.
VT (The lowest branch of the tassel is visible, but silks are not)	Tassel	One tassel was extracted from each corn plant
R1 (Silk is visible)	Silk	The ear was bagged, and the ear with the bag was removed from the plant and moved to a pollen-free environment, and the silk were cut from the ear.
R4 (Kernel contents are pasty as starch accumulates)	Grain	Thirty young grains were collected from each ear.

**Table 2 plants-11-03125-t002:** Source information of tested insects.

Insect Species	Date	Collecting Location	Insect Number Collected	Insect Stage	Host Plants
*S. frugiperda*	January 2019	Ruili City, Yunnan Province (23°58′ N, 97°48′ E)	150–200	3–5 instar larvae	Maize
*H. armigera*	August 2016	Yuanyang County, Henan Province (35°13′ N, 113°42′ E)	150–200	4–5 instar larvae	Maize
*O. furnacalis*	May to June 2015	Yuanyang County, Henan Province (35°13′ N, 113°42′ E)	150–200	4–5 instar larvae	Maize
*C. punctiferalis*	September 2016	Yuanyang County, Henan Province (35°13′ N, 113°42′ E)	250–300	3–5 instar larvae	Maize and sorghum
*M. separata*	August 2016	Lingbao County, Henan Province (34°36′ N, 110°48′ E)	300–350	4–5 instar larvae	Maize
*L. loreyi*	May 2019	Xinyang City, Henan Province (32°17′ N, 114°01′ E)	100–150	4–5 instar larvae	Maize
*A. lepigone*	May to June 2016	Yuanyang County, Henan Province (35°13′ N, 113°42′ E)	100–120	Adults	Captured by light trap

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
