# Peer review of "High-Dose Assessment of Transgenic Insect-Resistant Maize Events against Major Lepidopteran Pests in China"

_plants, 2022, doi:10.3390/plants11223125_

Round 1

Reviewer 1 Report

Lepidopteran pest is a key problem for crop production such as maize and cotton in most countries of the world. In order to develop new strategy for the pest control, GM technology has been used to develop crops with insect-resistant traits. However, some target insects can also develop resistance to GM insect-resistant crops,which has occurred in many countries.The high dose stragegy has been proved to be an effective pest-resistant management. The present manuscript described the content of the checking results of high-dose corn varieties to different target pests in China,which is very important for the food and environment safety.

1,Please provide information on the resistant development of target pests in the world,especially that of the relevant transgenes to be commercially planted in the GM corns in China.

2, Bt cotton has been planted commercially in China for more than twenty years, how to avoid the cross-resistant pest development with Bt corns?

3, Monitoring the insect-resistant develop is crucial and should be emphasized in the discussion section.

Author Response

  1. Please provide information on the resistant development of target pests in the world,especially that of the relevant transgenes to be commercially planted in the GM corns in China.

Response: Accepted. We added it in the introduction section.

Field resistance cases of S. frugiperda to TC1507 event expressing Cry1F and MON810 event expressing Cry1Ab corn have been reported in Puerto Rico, Argentina, and Brazil[7-9]. Busseola fusca in South Africa and Helicoverpa zea in the United States evolved resistance to MON810[10,11]. H. zea in the United States also presented resistance to MON89034 maize expressing Cry1A.105+Cry2Ab[12].

  1. Bt cotton has been planted commercially in China for more than twenty years, how to avoid the cross-resistant pest development with Bt corns?

Response: Accepted. We add it in the Discussion Section.

Because China has commercially planted Bt-Cry1Ac Cotton since 1997[39], one concern arises for cross- resistant pest development with Bt corns. This is not an important issue because more than 80 percent of cotton is grown in western China where is a non-corn-producing area. For mixed cotton and maize growing areas in eastern China, Bt-Cry1A maize with cross-resistance should be avoided to plant together in consideration of resistance management.

  1. Monitoring the insect-resistant develop is crucial and should be emphasized in the discussion section.

Response: Accepted. We put it into the Discussion section.

Resistance monitoring is the basis for resistance management of the target pests. The succeptible baselines and resistance allele frequencies of the major target pests in different ecological regions should be established before commercialization, and a regular program for resistance monitoring should be conducted after commercialization.

Reviewer 2 Report

A lot of research work was carried out in this study. Useful information was generated from the study.  

Minor comments:

It will help to carry out editorial reviews to help with smooth flow of some of the information. Some examples are shown below:

– Line 29. Change ‘does’ to ‘dose’

– Lines 288 to 291, Include ‘Ruifeng 125’ for the 4 average mortality rates shown.

– Line 356. “Absorbing foreign transgenic insect resistant corn…” the implication of the word ‘Absorbing’ is not clear.

– Page 14, line 384. “…and control of O. furnacalis is mainly, we should focus on breeding…”. I suggest changing to “…and control of O. furnacalis is the main goal, we should focus on breeding…”.

– Lines 423 to 425. I suggest changing to “For the future, further study is needed on the harm of the pests on corn as well as feeding characteristics in different regions of China’s corn belt.”

– Lines 446 and 447. Provide the names and brands of the dipsticks used.

– Change lines 483-484 to “Dilution and bioassay of lyophilized Bt maize tissues were performed using artificial high-dose assays [45, 46]

– Change 511-512 to “…events on four major lepidopteran pests in China.”

– Label the last reference as ‘46’.

Author Response

1– Line 29. Change ‘does’ to ‘dose’

Response: Done.

2– Lines 288 to 291, Include ‘Ruifeng 125’ for the 4 average mortality rates shown.

Response: We have added “Ruifeng 125 (Bt-Cry1Ab/Cry2Aj) and” in the sentense. 

3– Line 356. “Absorbing foreign transgenic insect resistant corn…” the implication of the word ‘Absorbing’ is not clear.

Response: We have changed “Absorbing” into “Learning from”.

4– Page 14, line 384. “…and control of O. furnacalis is mainly, we should focus on breeding…”. I suggest changing to “…and control of O. furnacalis is the main goal, we should focus on breeding…”.

Response: Done.

5– Lines 423 to 425. I suggest changing to “For the future, further study is needed on the harm of the pests on corn as well as feeding characteristics in different regions of China’s corn belt.”

Response: Accepted.

6– Lines 446 and 447. Provide the names and brands of the dipsticks used.

Response: Done. AA0331-LS for testing Cry1Ab and AA1632-LS for testing Vip3A (Shanghai Youlong Biotechnology Co., LTD)

7– Change lines 483-484 to “Dilution and bioassay of lyophilized Bt maize tissues were performed using artificial high-dose assays [45, 46].”

Response: Done.

8– Change 511-512 to “…events on four major lepidopteran pests in China.”

Response: Done. 

9– Label the last reference as ‘46’.

Response 9: Done.
